# Impaired Release of Neutrophil Extracellular Traps and Anemia-Associated T Cell Deficiency in Hereditary Hemorrhagic Telangiectasia

**DOI:** 10.3390/jcm9030767

**Published:** 2020-03-12

**Authors:** Freya Droege, Ekaterina Pylaeva, Elena Siakaeva, Sharareh Bordbari, Ilona Spyra, Kruthika Thangavelu, Carolin Lueb, Maksim Domnich, Stephan Lang, Urban Geisthoff, Jadwiga Jablonska

**Affiliations:** 1Department of Otorhinolaryngology, Head and Neck Surgery, Essen University Hospital, University Duisburg-Essen, Hufelandstrasse 55, 45147 Essen, Germany; Carolinlueb@googlemail.com (C.L.); stephan.lang@uk-essen.de (S.L.); 2Translational Oncology, Department of Otorhinolaryngology, University Hospital Essen, University Duisburg-Essen, Hufelandstrasse 55, 45147 Essen, Germany; ekaterina.pylaeva@uk-essen.de (E.P.); sharareh.bordbari@uk-essen.de (S.B.); ilona.spyra@uk-essen.de (I.S.); maksim.domnich@uk-essen.de (M.D.); jadwiga.jablonska@uk-essen.de (J.J.); 3Department of Otorhinolaryngology, Head and Neck Surgery, University Hospital of Marburg, University of Gießen and Marburg, Baldingerstrasse, 35042 Marburg, Germany; kruthika.thangavelu@gmail.com (K.T.); geisthof@med.uni-marburg.de (U.G.)

**Keywords:** Hereditary hemorrhagic telangiectasia, neutrophil extracellular traps, immune deficiency, cytoskeleton

## Abstract

Hereditary hemorrhagic telangiectasia (HHT) is characterized by mucocutaneous telangiectases and visceral vascular malformations. Individuals suffering from HHT have a significantly increased risk of bacterial infections, but the mechanisms involved in this are not clear. White blood cell subpopulations were estimated with flow cytometry in 79 patients with HHT and 45 healthy individuals, and association with clinicopathological status was assessed. A prominent decrease in absolute numbers of T cells in HHT was revealed (0.7 (0.5–1.1) vs. 1.3 (0.8–1.6), 10^6^/mL, *p* < 0.05), and in multivariate regression analysis, hemoglobin level was associated with lymphopenia (OR = 0.625, 95% CI: 0.417–0.937, *p* < 0.05). Although no changes in absolute numbers of neutrophils and monocytes were observed, we revealed a significant impairment of neutrophil antibacterial functions in HHT (*n* = 9), compared to healthy individuals (*n* = 7), in vitro. The release of neutrophil extracellular traps (NETs) against *Pseudomonas aeruginosa* MOI10 was significantly suppressed in HHT (mean area per cell, mm^2^: 76 (70–92) vs. 121 (97–128), *p* < 0.05), due to impaired filamentous actin organization (% of positive cells: 69 (59–77) vs. 92 (88–94), *p* < 0.05). To conclude, this study reveals the categories of patients with HHT that are prone to immunosuppression and require careful monitoring, and suggests a potential therapeutic strategy based on the functional activation of neutrophils.

## 1. Introduction

Hereditary hemorrhagic telangiectasia (HHT), also known as Rendu–Osler–Weber disease, is a rare, genetic, systemic disease characterized by mucocutaneous telangiectases and visceral vascular malformations. Patients show mutations in several genes in the transforming growth factor-beta/bone morphogenetic protein (TGFβ/BMP) signaling pathways and are often diagnosed with anemia due to recurrent epistaxis or gastrointestinal bleeding [1,2,3,4,5,6]. Clinically, an association between HHT and infectious diseases has been reported. Numerous cases indicated that particularly patients with HHT and pulmonary vascular malformations (PAVMs) or prolonged nasal packings for epistaxis are more susceptible to infections, cerebral abscesses, or septic thrombi [7,8,9,10,11]. A few studies reported life-threatening extracerebral severe infections, including osteoarthritis, septicemia, and spondylodiscitis, typically due to *Staphylococcus aureus* infection [12,13]. As PAVMs could cause paradoxical embolism of septic material or thrombi [14] preventing bacteremia, an antibiotic prophylaxis according to endocarditis guidelines is recommended for interventions with bacteremia [15].

Neutrophils are crucial innate immune cells that provide the first line of defense against bacterial pathogens due to their ability to rapidly migrate to the infectious focus and restrict bacterial dissemination to other organs [16]. Moreover, an impaired neutrophil functionality has been associated with an elevated susceptibility to bacterial infections [17]. Multiple mechanisms are involved in antibacterial activity of neutrophils, including release of reactive oxygen species (ROS), phagocytosis, secretion of antibacterial proteins, or the formation of neutrophil extracellular traps (NETs) [18].

Reactive oxygen species are critical components of the antimicrobial repertoire of neutrophils. The exact mechanisms involved in ROS-mediated killing of pathogens are not fully understood; however, ROS production is essential for killing of ingested pathogens, antigen presentation, or cross-presentation [19,20]. The formation of neutrophil extracellular traps (NETs) allows neutrophils to immobilize and kill invading bacteria with the cytotoxic granule peptides associated with decondensed DNA [21,22]. Insufficient ROS or NET formation can result in ineffective antibacterial defense [23]. 

These findings led us to hypothesize that patients with HHT might have an impaired neutrophil functionality and therefore be more susceptible to bacterial infections. As the experimental evidence on the regulation of immune responses in HHT patients is sparse and inconsistent, we assessed alterations of the immune system in such individuals, in comparison to healthy controls, to reveal factors predisposing such individuals to recurrent bacterial infections. 

## 2. Experimental Section

### 2.1. Selection of Participants and Clinical Characteristics 

In a prospective way, adult patients with HHT were diagnosed with the help of the Curaçao criteria or by genetic testing. Patients with HHT fulfilled at least three out of four Curaçao criteria and/or had a positive genetic testing [24,25]. Frequency, duration, intensity of epistaxis, and the Epistaxis Severity Score (ESS) for HHT [26] were used to quantify epistaxis. In addition, numerous clinical parameters including the need for medical attention, transfusions related to epistaxis, and signs of anemia (hemoglobin level) were documented. The clinical status including patients’ medical history, especially chronic inflammatory or immunosuppressive diseases, and patients’ smoking habits were recorded. Moreover, patients were asked to estimate the course of their disease and nosebleeds in the last four to six weeks. Patients could report deterioration, improvement, or no change.

For phenotypic and functional analyses of patient immune cells, blood samples were taken from the peripheral arm vein in citrate tubes (S-Monovette® 10 mL 9 NC, SARSTEDT AG & Co KG, Nümbrecht, Germany). All experiments were also performed with blood from a control group. The number of patients and healthy donors is listed for each experiment. At enrollment in the study, no patient or control person suffered from an acute infectious/inflammatory disease.

### 2.2. Laboratory and Immunological Parameters of the Whole Blood

The phenotypic characterization of immune cell subsets was performed using whole blood and standard immunofluorescence and flow cytometry technology. Main cell populations (lymphocytes, monocytes, neutrophils, and eosinophils) were assessed with morphological gating and according to surface markers’ expression. The following monoclonal antibodies were used for staining: anti-CD3 (clone BW264/56, Miltenyi Biotec, Bergisch Gladbach, Germany), anti-CD8 (clone SK1, BioLegend, San Diego, CA, USA), anti-CD56 (clone B159, BD Biosciences, BD, New Jersey, NY, USA), anti-CD45RO (clone UCHL1, Miltenyi Biotec, Bergisch Gladbach, Germany), anti-CD45RA (clone HI100, BioLegend, San Diego, CA, USA), anti-CD62L (clone DREG-56, BD Biosciences, BD, Franklin Lakes, U.S.), and anti-CD279 (PD1) (clone eBioJ105, eBioscience, Thermo Fisher Scientific, Waltham, U.S.) with isotype control (clone MOPC-21, BioLegend, San Diego, CA, USA). Cell counts were assessed using the BD FACS Canto system, and data were analyzed using BD FACS Diva software (BD Biosciences, BD, Franklin Lakes, U.S.). The gating strategy is depicted in Figure 1A. The percentage of the cells and absolute numbers (×10^6^/mL) were calculated. Lymphocyte numbers lower than 1.0/nL were coded as lymphocytopenia [27]. Women with an average hemoglobin level below 12.0 g/dl and men with a level below 13.0 g/dL were classified as having anemia [28].

### 2.3. Plasma

Citrate plasma was collected from each blood sample after centrifugation and stored at −80 °C.

### 2.4. Isolation of Human Neutrophils

Peripheral blood was drawn into 3.8% sodium citrate anticoagulant monovettes, and separated by density gradient centrifugation (Biocoll density 1.077 g/mL, Biochrom, Merck, Darmstadt, Germany). The mononuclear cell fraction was discarded, and neutrophils (purity >95%) were isolated by sedimentation over 1% polyvinyl alcohol, followed by hypotonic lysis (0.2% NaCl) of erythrocytes. In view of the emerging diversity of circulating neutrophil subtypes in humans, note that high-density neutrophils have been investigated in this study.

### 2.5. Bacteria 

*Pseudomonas (P.) aeruginosa* PA14 strain (wild-type serogroup O10 strain, cytotoxic ExoU+) was used in the study. Bacteria were cultured in Luria–Bertani (LB) broth for the 3 h to reach the early exponential phase, washed twice in PBS (Gibco, Thermo Fisher Scientific, Waltham, U.S.), and the optical density of 100 µL suspensions was measured in 96-well flat-bottom cell culture plates (Cellstar, Greiner Bio One International GmbH, Frickenhausen, Germany) at 600 nm using the microplate reader Synergy 2 (BioTek Instruments, Inc., Vermont, U.S.). OD 0.4 corresponds to a bacterial density of 5 × 10^9^/mL, as determined by serial dilutions and colony forming unit (CFU) assays. Bacteria concentration was adjusted to the desired values and verified by plating on 2% LB agar plates. 

### 2.6. NET Release

Isolated neutrophils 25,000/well were incubated with *P. aeruginosa* (MOI 10) in a glass-bottom 96-well plate (MatTek Corporation, Ashland, U.S.) pre-coated with poly-D-lysine 1 mg/mL (Sigma-Aldrich/Merck, Darmstadt, Germany) for 4 h at +37 C, 5% CO_2_; sterile medium Roswell Park Memorial Institute (RPMI, Gibco, Thermo Fisher Scientific, Waltham, U.S.) containing 10% fetal calf serum (FCS, Biochrom, Merck, Darmstadt, Germany) was used as a negative control. Samples were fixed with paraformaldehyde (Thermo Fisher Scientific, Waltham, U.S.) to the final concentration of 4%, permeabilized with Triton X-100 (Sigma-Aldrich/Merck, Darmstadt, Germany) 0.2% containing buffer, stained with DAPI (4’,6-diamidino-2-phenylindole, BioLegend, San Diego, CA, USA), anti-histone 1 (Merck Millipore, Darmstadt, Germany), and donkey-anti-mouse-AF564 (Invitrogen, Thermo Fisher Scientific, Waltham, U.S.) secondary antibodies, and mounted with ProLong Gold Antifade Mountant with DAPI (Invitrogen, Thermo Fisher Scientific, Waltham, U.S.). The mean area of NETs (extracellular histone 1) released by NET-positive neutrophils was estimated.

### 2.7. Reactive Oxygen Species

Isolated neutrophils in amounts of 100,000/well were washed and resuspended in RPMI containing 10% FCS, and *P. aeruginosa* MOI 10 was added. Sterile medium was used as a negative control. ROS production by isolated neutrophils was estimated after 60 min of exposure to *P. aeruginosa* using Dihydrorhodamine 123 (Sigma-Aldrich/Merck, Darmstadt, Germany) with flow cytometry. 

### 2.8. F-Actin Detection with Phalloidin

Isolated neutrophils in amounts of 100,000/well were washed and resuspended in RPMI containing 10% FCS, *P. aeruginosa* MOI 10, or phorbol 12-myristate 13-acetate (PMA, Biomol, Hamburg, Germany) in sterile tubes (Corning, Falcon, New York, NY, U.S.) for flow cytometry or in a glass-bottom 96-well plate (MatTek Corporation, Ashland, MA, U.S.) pre-coated with poly-D-lysine 1 mg/mL (Sigma-Aldrich/Merck, Darmstadt, Germany) for microscopy for 15 minutes at +37 C; 5% CO_2_; sterile medium was used as a negative control. Samples were fixed with paraformaldehyde (Thermo Fisher Scientific, Waltham, MA, U.S.) to the final concentration of 4%, permeabilized with Triton X-100 (Sigma-Aldrich/Merck, Darmstadt, Germany) 0.2% containing buffer, stained with DAPI (4’,6-diamidino-2-phenylindole, BioLegend, San Diego, CA, USA) and phalloidin (Flash Phalloidin^TM^ Green 488, BioLegend, San Diego, CA, USA), and mounted with ProLong Gold Antifade Mountant with DAPI (Invitrogen, Thermo Fisher Scientific, Waltham, MA, U.S.). Percent of phalloidin-positive cells and mean fluorescence intensity were estimated with flow cytometry, and the reorganization of the actin cytoskeleton was assessed microscopically. 

### 2.9. Microscopy

Microscopy was performed using a Zeiss AxioObserver.Z1 inverted microscope with ApoTome optical sectioning and a Zeiss ELYRA PS.1 super-resolution microscope with Structured Illumination (SIM), Pal-M/STORM localization and Total Internal Reflection Fluorescence (TIRF), combined with an LSM710 Confocal equipped with filters for DAPI, fluorescein (FITC), and Alexa Fluor 564. Images were processed with ZEN Blue 2012 software (all from Carl Zeiss, Oberkochen, Germany) and analyzed with ImageJ (version 1.52i, National Institutes of Health, Bethesda, MD, U.S.). 

### 2.10. Migration to Lipopolysaccharide (LPS)

Cell migration was evaluated using a two-chamber transwell system (3 μm pore size) with cell culture inserts (Corning, Falcon, New York, U.S.). Dulbecco’s Modified Eagle’s Medium (DMEM) supplemented with 10% FCS containing LPS (Invitrogen, Thermo Fisher Scientific, Waltham, MA, U.S.) at a concentration of 1 ng/mL or LPS-free medium, as a negative control for spontaneous migration, were added to the lower chamber. Isolated neutrophils from healthy individuals and HHT patients, 1 × 10^6^ cells in DMEM with 10% FCS, were added to the upper chamber and placed into medium for 3 h at 37 °C and 5% CO_2_. Cells that had transmigrated to the lower chamber were counted using CASY (Roche Innovatis AG, Bielefeld, Deutschland). The chemotaxis index was determined using the following formula: Chemotaxis index = (Stimulated–Spontaneous) * 100%/Spontaneous.

### 2.11. ELISA

ELISA for plasma total TGFß levels was performed using a Human TGF-beta 1 DuoSet ELISA kit (R&D systems, Bio-Techne Ltd., Abingdon, UK) according to the manufacturer’s protocol.

### 2.12. Statistical Analysis

Normality of distribution was checked with the Kolmogorov–Smirnov test. Descriptive statistics (number, percentage of patients (N, %) and mean ± standard deviation (m ± SD) for normally distributed parameters or mean, interquartile range, minimal and maximal values for non-normal distribution) were used for patients’ clinical presentations. Normally distributed samples were compared using ANOVA for multiple comparison and Student *t*-tests for two independent samples; for nonparametrically distributed samples, Kruskal–Wallis ANOVA with the Bonferroni correction for multiple comparisons and the Mann–Whitney U test for two independent samples, chi-square test (Χ^2^), and odds ratio (OR) with 95% confidence interval (95% CI) were used. To measure the predictor variables, a multivariate regression analysis was performed. Correlations were estimated with the Spearman rank R test, and comparison of correlations was done using Fisher’s r to z transformation and z test.

### 2.13. Study Approval

The protocol was approved by the Ethics Committee of the University Duisburg-Essen (16-7026-BO). All subjects gave written informed consent in accordance with the Declaration of Helsinki. The study was registered at Clinical trials.gov (ID NCT02983253).

## 3. Results

### 3.1. Clinical Characteristics of the HHT Patients

The study included 44 female and 35 male HHT patients with an average age of 55 years (SD ± 13 years). Genetic testing was performed in 22% of the patients (n = 17/79), with seven patients suffering from HHT 1 (n = 7/17, 41%) and five patients from HHT 2 (n = 5/17, 29%). In one patient, no typical HHT mutation was found (n = 1/17, 6%), and in four cases, no specific results were available (n = 4/17, 24%). Most patients presented multiple typical telangiectasia (n = 77/79, 98%) and recurrent epistaxis (n = 77/79, 98%, Epistaxis Severity Score (ESS) (m ± SD): 6 ± 2 on a scale from 1 to 10). There was a positive correlation between patients’ age and the ESS (r = 0.303, *p* = 0.018), and older patients suffered more often from gastrointestinal involvement (*t*-test = 2.404, *p* = 0.020). Comparing 79 patients with HHT and 45 healthy controls, no significant changes in sex and smoking habits were documented (sex: Χ^2^ = 0.436, *p* = 0.509, smoking habits: Χ^2^ = 4.185, *p* = 0.123) (Table 1). 

All patients fulfilled at least three out of four Curaçao criteria (n = 79) and/or had a positive genetic testing (n = 13). Data are presented as mean ± standard deviation or n (%) (percentages are calculated after exclusion of missing values).

### 3.2. Immune Status of Healthy and HHT Individuals of Comparable Age 

We aimed to study the immune parameters in HHT to reveal factors that could be responsible for the immunosuppression and higher infection susceptibility reported for these patients. For this purpose, we collected blood from HHT patients and healthy individuals (Table 2). There was a significant difference in the mean age of HHT patients and controls (*t*-test: −6.044, *p* < 0.001). For further analysis, we used data obtained from an age-matched subgroup of HHT (n = 58/79, 73%) and healthy (n = 23/45, 51%) individuals (Table 2). Reduced absolute numbers of lymphocytes, mainly CD4+ T cells (Table 2, Figure 1B), were observed in HHT patients, as compared to healthy individuals. This led to a relative predominance of neutrophils and monocytes in HHT patients (Figure 1C,D). Within T cells of HHT patients, a strong reduction of naive cells and predominance of central memory T cells was observed (Figure 1E). Among CD4+ lymphocytes, the proportion of PD1-positive cells was higher in HHT (Figure 1F). No differences in plasma TGFß levels were observed between groups (Table 2).

### 3.3. Age Is a Significant Factor Contributing to Declined Immune Parameters in HHT

Taking into consideration the possible role of age in observed changes in the immune system, we performed an analysis of the correlation between age and immune parameters. We could observe different patterns of age-related alterations of the immune cell numbers between HHT patients and healthy individuals. Particularly, although in healthy individuals the total lymphocytes and T cell counts were not changed, the HHT group showed a significant decline of these parameters with age. Moreover, relative levels of monocytes increased with age in HHT, but not in healthy individuals (Figure 2A). Importantly, a prominent decrease in red blood cells was also observed in the HHT group (Figure 2B).

### 3.4. Anemia Is Associated with a Significant Decrease in Lymphocytes Counts in HHT

As blood loss may contribute not only to anemia, but also to the loss of immune cells, we estimated the correlations between anemia and immune parameters. A significant positive correlation of whole blood cell parameters, especially erythrocytes counts, hemoglobin levels, and hematocrit, and cells of adaptive immunity (absolute numbers of T and B lymphocytes) was observed (Table 3).

Accordingly, negative correlations between hemoglobin level and ESS total score (Figure 3A), and between lymphocyte counts and ESS total score (Figure 3B,C) were detected. This indicates that blood loss in HHT due to epistaxis results mainly in loss of lymphocytes, whereas myelopoiesis is relatively preserved in HHT. 

In multivariate regression analysis, the hemoglobin level was associated with lymphocytopenia (hemoglobin level: odds ratio (OR) = 0.625, 95% confidence interval (CI): 0.417–0.937, *p* = 0.023; ESS: OR = 1.350, 95% CI: 0.831–2.193, *p* = 0.226; age: OR = 1.033, 95% CI: 0.960–1.112, *p* = 0.387), but not with self-estimation of bleeding severity (see supplementary materials, Appendix A). As iron replacement treatment may influence immune parameters [29], we analyzed the immune cell levels in groups of HHT under different therapies (no therapy due to lack of indications, oral iron, and intravenous +/− oral iron). Although red blood parameters varied significantly between groups, with the lowest values in the intravenous +/− oral iron group, no differences in absolute numbers of immune cells in blood between groups were observed (Table 4).

### 3.5. Immune Parameters Correlate with Type of Vascular Malformation in HHT

HHT patients with PAVM are known to suffer from a high rate of bacterial complications, specifically abscesses and strokes due to the loss of the pulmonary capillary filter by the pulmonary shunt [14]. Here, we could observe that one fifth of the patients (n = 18/79, 23%) reported a chronic inflammatory or immunosuppressive disease or took an immunosuppressive medication (CIDT) (Table 1). Patients suffering from gastrointestinal bleeding (GI) and those with a lower hemoglobin level tend to be more often diagnosed with a CIDT (GI: Χ^2^ = 5.277, *p* = 0.022; hemoglobin: *t*-test = 1.686, *p* = 0.097). Six patients (n = 6/79, 8%) reported having had an abscess (Table 1). Patients with PAVM did not show a higher rate of CIDT (Χ^2^ = 0.244, *p* = 0.621) but were more prone to develop an abscess than those without these vascular malformations (PAVM: OR = 4.000, 95% CI: 0.665–24.065, *p* = 0.109; HVM: OR = 1.577, 95% CI: 1.250–1.990, *p* = 0.099; CVM: OR = 1.140, 95% CI: 1.026–1.265, *p* = 0.407; GI: OR = 1.625, 95% CI: 0.250–10.578, *p* = 0.609). Therefore, in accordance with other studies, we observed that the presence of PAVM led to a higher risk of developing abscesses.

### 3.6. Reduced Functionality of Neutrophils Isolated From Patients with HHT

As all the above results could not fully explain the elevated susceptibility of HHT individuals to infections, we decided to evaluate the antimicrobial activity of neutrophils in these patients. We evaluated neutrophils from nine HHT patients and seven healthy controls (Table 5). No significant changes in age and sex were documented between these groups (age: *p* = 0.87, Mann–Whitney U test; sex: Χ^2^ = 0.12, *p* = 0.73).

First, we assessed ROS production by neutrophils isolated from HHT patients in response to bacterial stimulation and compared it to healthy individuals. We exposed isolated neutrophils for 1 h with *P. aeruginosa* (multiplicity of infections (MOI) 10) or in control, and measured ROS release after 1 h. We could not observe any significant difference in ROS release between healthy and HHT neutrophils (Figure 4A).

Next, we evaluated bacteria that triggered NETosis in neutrophils isolated from HHT and healthy donors. For this, we challenged isolated neutrophils with *P. aeruginosa* MOI 10 for 4 h, using sterile medium as a control. Notably, we could observe significantly impaired NET release in neutrophils isolated from HHT patients (Figure 4B,C). Even though the spontaneous NET release was not reduced in HHT, NETosis in response to *P. aeruginosa* infection was significantly diminished in such patients. Possibly, this phenomenon is responsible for the observed immune suppression of HHT patients.

### 3.7. Altered Cytoskeleton in Neutrophils Isolated From HHT Patients

The cellular events driving NETosis are still unclear. However, the cytoskeleton seems to be involved in this process [30]. To assess cytoskeleton differences between healthy and HHT individuals, we performed phalloidin staining of isolated blood neutrophils challenged with phorbol-12-myristat-13-acetat (PMA) or bacteria (*P. aeruginosa*) for 15 minutes; sterile medium was used as a control. 

We observed significantly lower amounts of F-actin in HHT neutrophils in steady state, as well as after stimulation (Figure 5A). 

To evaluate structural changes in the cytoskeleton of HHT neutrophils, we performed microscopical examination of these cells after stimulation, as described above. Here, we also observed lower levels of F-actin in neutrophils isolated from HHT patients (Figure 5B). Moreover, we noticed signs of impaired cytoskeleton organization in response to the stimulation in these neutrophils (Figure 5C,D). The area of adhesion to the surface was significantly lower in HHT neutrophils, especially after stimulation with *P. aeruginosa*, as compared to healthy neutrophilis (Figure 5D). Concurrently, the size of the cells in solution, measured by flow cytometry [31], did not differ between HHT and healthy groups under all conditions (Figure 5C). As the adhesive interactions play a central role in neutrophil transmigration, we assessed the motility of HHT neutrophils and compared it to the motility of healthy neutrophils. Indeed, the decreased adhesion was associated with enhanced motility to lipopolysaccharide (LPS) in HHT cells (Figure 5E).

Thus, the impaired cytoskeleton reorganization observed in HHT neutrophils influences their vital and antibacterial functions, such as NET release, adherence, and migration, and may contribute to the observed immunodeficiency in patients suffering from HHT.

## 4. Discussion

HHT is a disease characterized mainly by mutations in endoglin (*ENG*, HHT type 1) or activin receptor-like kinase 1 (*ALK1*, HHT type 2). It is associated with a range of systemic abnormalities such as mucocutaneous telangiectasia—the reason for bleeding and anemia—and arteriovenous malformations in organs, which cause blood shunting and hemodynamic failure [32,33]. Moreover, there is evidence that HHT patients are more prone to develop certain infectious complications [7,11,12,13]. The majority of infections in HHT patients are bacterial, caused by *Actinomyces*, *Fusobacterium*, *Streptococcus*, and anaerobic species of *Streptococcus*. Transient bacteremia in such patients (due to surgical procedures, intravenous line placement, or dental problems) may result in serious complications such as abscesses of the brain or other organs [34]. Moreover, HHT patients with prolonged epistaxis episodes have an increased susceptibility to *Staphylococcus aureus* bacteremia and its seeding to various sites (such as vertebrae or joints) [11].

HHT patients with PAVMs are known to suffer from a high rate of bacterial complications, abscesses, and strokes due to the loss of the pulmonary capillary filter by the existing right to left shunt through the PAVM [11,14,35]. In accordance with other studies, we observed that PAVM led to a higher risk of developing abscesses.

The observed elevated susceptibility of HHT patients to develop bacterial complications was the rationale to investigate the potential mechanism in this study. We have evaluated the changes in innate and adaptive immunity of HHT patients and correlated them with clinical parameters.

The analysis of main white blood cell (WBC) populations and further investigation of lymphocyte subpopulations revealed a decrease in absolute numbers of T cells, especially in CD4+, which leads to the relative predominance of B and NK lymphocytes, neutrophils, and monocytes in blood. In accordance with this, other studies showed decreased CD4+ T cell levels in patients with HHT [36,37]. In response to infections, an increased number of central memory T cells can be detected as they accumulate over the lifetime [38]. Accordingly, higher numbers of central memory T cells in HHT patients compared to healthy individuals were observed, possibly indicating that HHT patients suffer more often from infections than controls, while also possibly being an indirect sign of preserved adaptive immune mechanisms.

The onset of HHT and the epistaxis severity are variable for each individual patient, and the severity intensifies with increasing age [39]. In accordance with this, there was a positive correlation between age and patients’ ESS. We observed that age is a significant factor contributing to decreased immune parameters in HHT. In agreement with this, Lenato et al. showed a decline in immunological parameters in aged HHT patients, whereas younger individuals were comparable with healthy controls [40]. The correlation analysis performed in the present study has proven the continuous impairment of immune parameters (lymphocyte numbers and relative redistribution of cell populations) with age in HHT. This was not observed in healthy individuals. Our finding of the higher frequency of PD1+ CD4+ cells in HHT may reflect the immunosenescence and exhaustion of the immune system [41]. This could lead to the impaired functions of T lymphocytes and result in the higher burden of infections in HHT. In agreement with this, the imbalance of inflammatory and anti-inflammatory networks with aging (“inflammaging”) has already been described [42,43,44]. Changes in immunity during normal aging include the activation of the innate and adaptive immune responses, which, in the long run, leads to an exhaustion of the adaptive immune system [42]. Accumulated memory cells occupy the immunological space, repressing the production of naive T cells [45,46]. A potential influence, which cannot be excluded totally, is how far former recurrent infections have contributed to exhausted T lymphocytes in the elderly.

Clinical studies in patients with chronic iron deficiency due to conditions other than HHT showed differences regarding the humoral immune system and nonspecific immunity [47,48]. In our study, a significant connection between deficiency of the adaptive immunity and severity of iron deficiency and anemia could be shown. Although neutrophils and monocytes are rapidly substituted from progenitor cells, lymphocyte progenitors proliferate slower [49], which might lead to the deficiency in absolute T cells numbers during the chronic loss of lymphocytes. This phenomenon highlights the role of uncorrected blood loss in the impairment of the immune system in HHT patients. However, *ENG*^-/-^ mice that are used as a model for HHT do not show the impaired lymphocyte status [50]. Those mice also do not develop anemia, but are more susceptible to bacterial infections due to impairment in the innate immunity. To our knowledge, the chronic blood loss in individuals suffering from HHT has not yet been simulated in a murine model. Therefore, translation of the results from HHT animal studies into the human situation should be done carefully.

An association between iron administration as a treatment for anemia, immune deficiency, and predisposition to bacterial infection was hypothesized; namely, a high and prolonged intake of iron, as is often the case in patients with HHT suffering from chronic iron deficiency anemia, could increase the level of oxidative stress, DNA-damage, and lymphopenia [51,52,53]. However, we did not observe any association between iron administration (oral and/or intravenous) and patients’ immune status.

Besides the quantitative loss of immune cells, several functional defects of the immune system in HHT such as a reduction in polymorphonuclear cells and monocyte functions (reduced phagocytosis and oxidative burst) in HHT individuals have been described [54]. As neutrophils play a major effector role during bacterial infections, we addressed the alterations in the activity and functionality of these cells in HHT patients as compared to healthy individuals. Neutrophils exert their antibacterial functions via ROS and NET release. Although ROS production, which can serve as an additional trigger for NETosis [55], was not significantly changed between the HHT and healthy cohorts, NET production in response to *P. aeruginosa* was significantly decreased in patients with HHT. We observed a reduction of almost 50% in NET area in HHT, suggesting impaired antibacterial properties of neutrophils in this disease. The crucial role of NETosis in antibacterial immunity is demonstrated in studies on chronic granulomatous disease. This genetic disorder is characterized by dramatic decreases in ROS and NET production due to nicotinamide adenine dinucleotide phosphate (NADPH) oxidase defects. As a result, such patients are predisposed to severe life-threatening bacterial and fungal infections [56].

The observed defect in NET release could be due to the impairment of cytoskeleton reorganization, as it is known that F-actin participates in this process [30]. Indeed, we could observe, for the first time, the impaired polymerization of actin fibers in neutrophils of HHT patients after stimulation with live bacteria or PMA. A disorganized cytoskeleton has already been described in circulating endothelial cells. Decreased endoglin expression due to mutations in both *ENG* and *ALK1* genes was suggested to play a role in this process [57]. The impaired interaction between endoglin and zyxin-related protein 1 [58] may play a role in diminished cytoskeleton reorganization in endothelial cells. Although *ENG* and *ALK1* are considered to be predominantly expressed in endothelial cells [59], some authors have shown the expression of both of these genes in blood monocytes or lymphocytes [60,61] as well as in hematopoietic stem cells [62]. Thus, it is possible that cytoskeleton organization in neutrophils also depends on these proteins, making these cells vulnerable to their loss in HHT.

Alterations in the actin organization of neutrophils do not only affect NETosis, but also other vital functions of neutrophils, such as adhesion, migration, or degranulation. Here, we observed a significantly decreased area of cell spread on the poly-D-lysine-coated surface upon stimulation with *P. aeruginosa* in HHT. This also correlated with increased motility of the cells towards LPS. As cell–matrix adhesions represent the focal sites of convergence between the actin cytoskeleton and extracellular matrix [63], the appearance of focal adhesions may retard cellular movement. In agreement, a reverse correlation between cell spreading on the extracellular matrix or poly-d-lysine and motility was described [64]. However, unimpaired firm adhesion is important for transmigration processes [65].

### Study Limitations

Data about genetic testing were not available for all patients with HHT. However, we see homogenous changes in adaptive immunity in comparison to healthy volunteers, which may indicate similar mechanisms in both major mutations (HHT types 1 and 2). In rare diseases, small study populations are often found. In this study, 79 patients with HHT were evaluated. However, the altered actin cytoskeleton and NET release were analyzed in just nine patients with HHT and seven healthy controls. As cytoskeleton reorganization is involved in multiple functions of neutrophils, different steps of cytoskeleton-dependent neutrophil responses should be analyzed in more detail. Further investigations and correlations with genetic mutations are needed to fully understand immunologic changes in patients with HHT to address new therapeutic sites. 

## 5. Conclusions

In summary, we demonstrated both a quantitative decrease in lymphocytes in absolute numbers and qualitative impairments of neutrophils, including decreased capacity to produce NETs, and impaired cell adhesion and migration. Our data support the hypothesis that the reduced neutrophil functions are caused by impaired cytoskeleton reorganization. The changes described above are probably responsible for an impaired innate and adaptive immunity in HHT patients, clinically apparent as an elevated susceptibility to bacterial infections. Our data imply that patients with HHT might be at a higher risk of developing infectious diseases, which would have to be taken into clinical consideration. The observed altered parameters and improved understanding of the role of the cytoskeleton in NET formations can potentially lead to the development of novel therapeutic strategies to restore neutrophil antibacterial functions.

## Figures and Tables

**Figure 1 jcm-09-00767-f001:**
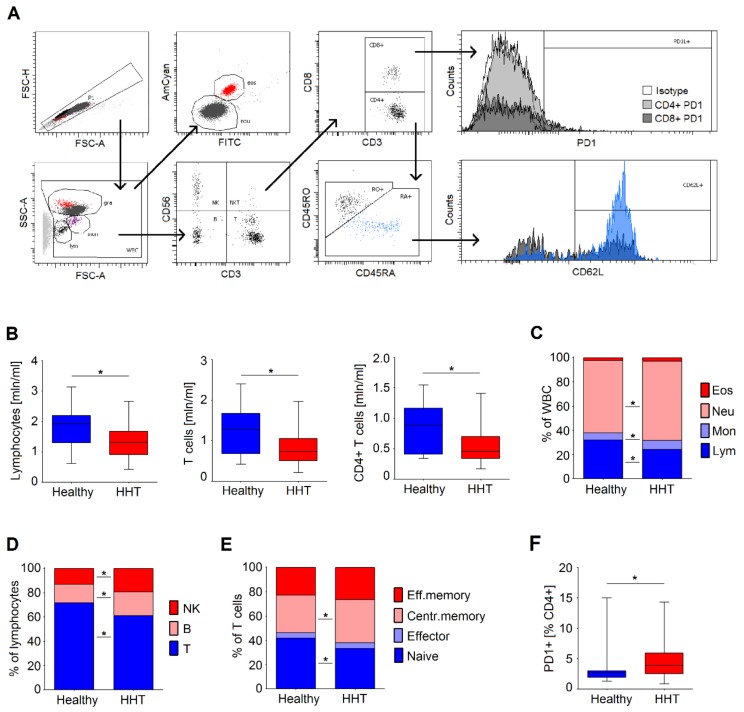
White blood cell populations in hereditary hemorrhagic telangiectasia (HHT) compared to healthy controls. (**A**) Gating strategy. Blood cell populations (lym: lymphocytes, mon: monocytes, gra: granulocytes, neu: neutrophils, eos: eosinophils) were assessed in whole blood after staining and erythrocyte lysis based on the morphological properties, and subpopulations of lymphocytes (T, B, NK cells, CD4+, CD8+, naive CD45RA+CD45RO-CD62Lhigh, effector CD45RA+CD45RO-CD62Llow, effector memory CD45RA-CD45RO+CD62Llow, central memory CD45RA-CD45RO+CD62Lhigh, exhausted PD1+) were determined based on the surface markers. (**B**) Decline in the absolute numbers of lymphocytes, T cells, and CD4+ T cells in HHT. Data are presented as median, 25–75 percentiles, and minimal–maximal values; Mann–Whitney U test was used to compare groups, * *p* < 0.05. (**C**) Relative predominance of neutrophils and monocytes in HHT due to decreased numbers of lymphocytes. (**D**) Relative predominance of B and NK cells in HHT due to decreased numbers of lymphocytes. (**E**) Increase in central memory T cells and decrease in naive T cells in HHT. Data are presented as median levels; Mann–Whitney U test was used to compare groups, * *p* < 0.05. (**F**) Increased proportion of PD1+ CD4+ T cells in HHT in comparison to healthy controls. Data are presented as median, 25–75 percentiles, and minimal–maximal values; Mann–Whitney U test was used to compare groups, * *p* < 0.05. HHT = hereditary hemorrhagic telangiectasia.

**Figure 2 jcm-09-00767-f002:**
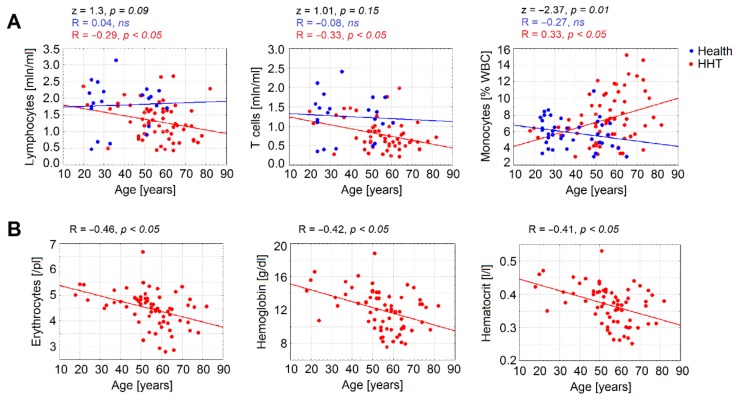
Age-related decline in immune and red blood parameters in HHT. (**A**) Age-dependent amounts of monocytes, lymphocytes, and T cells were analyzed in patients with hereditary hemorrhagic telangiectasia (HHT) compared to healthy controls. (**B**) Absolute number of erythrocytes, hemoglobin, and hematocrit decreased in patients with HHT. Data are presented as individual levels; Spearman R test was used to assess the correlations, and comparison of correlations was performed using Fisher’s r to z transformation and z test. Healthy: number of persons (n) = 23, HHT: n = 40, HHT = hereditary hemorrhagic telangiectasia.

**Figure 3 jcm-09-00767-f003:**
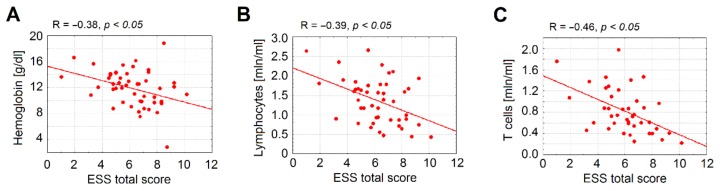
Negative correlation of the epistaxis severity and immune parameters. In patients with HHT and higher Epistaxis Severity Score (ESS), lower hemoglobin (**A**), lymphocyte (**B**), and T cell (**C**) levels were documented. Number of HHT patients = 40, HHT = hereditary hemorrhagic telangiectasia, ESS = Epistaxis Severity Score (on a scale from 1 = mild epistaxis to 10 = severe epistaxis).

**Figure 4 jcm-09-00767-f004:**
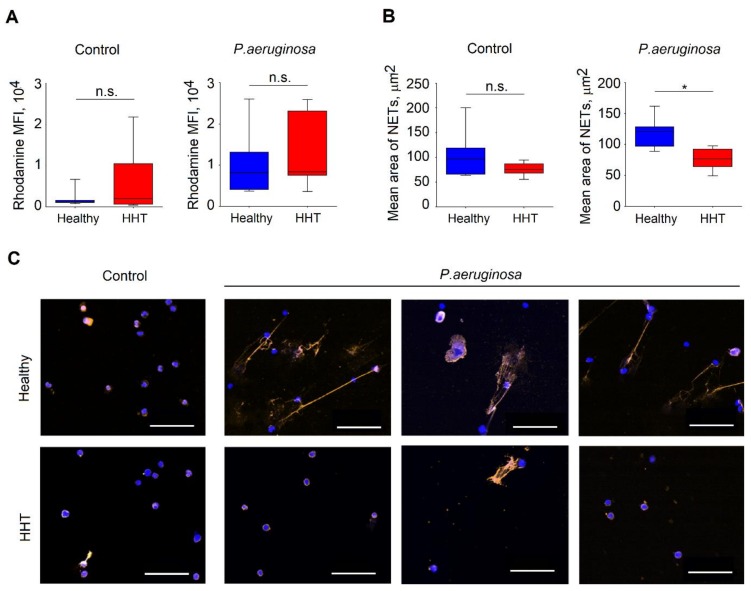
ROS and NET production in patients with HHT compared to healthy individuals. (**A**) ROS were detected in polymorphonuclear leucocytes (PMNs) of HHT patients and healthy individuals after challenge with *P. aeruginosa* and in control; no difference was observed. (**B**) Decreased mean area per one neutrophil covered with NETs in HHT in the presence of *P. aeruginosa.* Data are presented as median (25–75 percentiles); Mann–Whitney U test was used to compare groups, * *p* < 0.05. (**C**) Representative pictures illustrating the decreased NET formation in HHT. Histone 1 in orange, DAPI in blue, scale bar: 50 µm. Healthy: n = 7, HHT: n = 9. HHT = hereditary hemorrhagic telangiectasia, MFI = mean fluorescence intensity, ROS = reactive oxygen species, NETs = neutrophil extracellular traps, *P. aeruginosa* = *Pseudomonas aeruginosa*, 4′,6-diamidin-2-phenylindol = DAPI, h = hours, n = number of persons.

**Figure 5 jcm-09-00767-f005:**
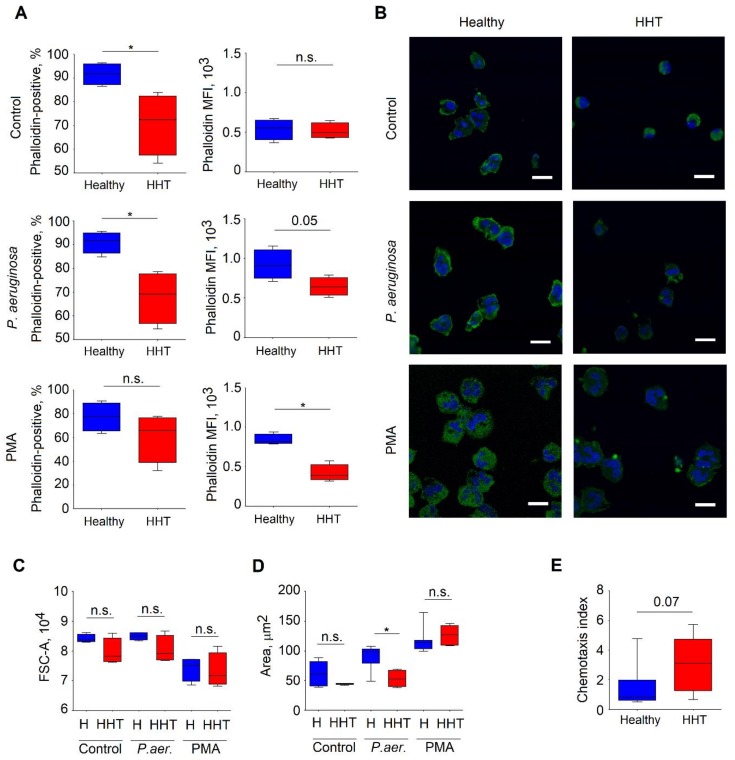
Altered actin cytoskeleton in HHT patients. (**A**) The percentage of phalloidin-stained neutrophils and mean fluorescence intensity (MFI) under different conditions (control, PMA and *P. aeruginosa*) was lower in HHT in comparison to healthy individuals. (**B**) Representative pictures illustrating altered actin cytoskeleton of neutrophils in HHT patients after stimulation with PMA or *P. aeruginosa*, DAPI in blue, F-actin (phalloidin) in green, scale bar: 10 µm. (**C**) No difference in cell size (proportional to FSC value) of neutrophils in suspension was observed between HHT and healthy groups. Comparison of the mean value of FSC of the neutrophils. (**D**) Smaller area of adherence of the neutrophils in HHT versus the healthy group after stimulation with *P. aeruginosa*. (**E**) Neutrophils in HHT in comparison to healthy probands had a tendency of better migratory capacity towards lipopolysaccharide. Data are presented as median (25–75 percentiles); Mann–Whitney U test was used to compare groups, * *p* <0.05. Healthy: n = 7, HHT: n = 9, HHT = hereditary hemorrhagic telangiectasia, MFI = mean fluorescence intensity, FITC = fluorescein, *P. aeruginosa* = *Pseudomonas aeruginosa*, PMA = phorbol-12-myristat-13-acetat, DAPI = 4′,6-diamidin-2-phenylindol, FSC = forward scatter, n = number of persons.

**Table 1 jcm-09-00767-t001:** Clinical manifestations of 79 patients with HHT.

		All Patients (n = 79)		Men (n = 35)	Women (n = 44)
**age**		years	55 ± 13	56 ± 14	54 ± 13
		missing	0	0	0
**genetic testing**		yes	17 (22)	6 (17)	11 (25)
		no	62 (78)	29 (83)	33 (75)
		missing	0	0	0
**genetic mutation**		HHT 1	7 (54)	1 (33)	6 (60)
		HHT 2	5 (39)	2 (67)	3 (30)
		no mutation detected	1 (8)	0	1 (10)
		missing	59	32	34
**positive FH**		yes	68 (92)	30 (91)	38 (93)
		no	6 (8)	3 (9)	3 (7)
		missing	5	2	3
**TAE**		yes	71 (98)	34 (97)	43 (98)
		no	2 (2)	1 (3)	1 (2)
		missing	0	0	0
**visceral lesions**	**GI**	yes	27 (49)	12 (50)	15 (48)
		no	28 (51)	12 (50)	16 (52)
		missing	24	11	13
	**PAVM**	yes	21 (37)	7 (30)	14 (41)
		no	36 (63)	16 (70)	20 (59)
		missing	22	12	10
	**HVM**	yes	15 (33)	4 (20)	11 (42)
		no	31 (67)	16 (80)	15 (58)
		missing	33	15	18
	**CVM**	yes	6 (11)	3 (13)	3 (10)
		no	48 (89)	20 (87)	28 (90)
		missing	25	12	13
**epistaxis**		yes	77 (98)	34 (97)	43 (98)
		no	2 (2)	1 (3)	1 (2)
		missing	0	0	0
**ESS**		score	6 ± 2	6 ± 2	6 ± 2
		missing	18	6	12
**course of epistaxis**		worse	57 (75)	24 (69)	33 (81)
		equal	10 (13)	5 (14)	5 (12)
		better	9 (12)	6 (17)	3 (7)
		missing	3	0	3
**course of disease**		worse	61 (81)	27 (82)	34 (81)
		equal	12 (16)	4 (12)	8 (19)
		better	2 (3)	2 (6)	0 (0)
		missing	4	2	2
**iron intake**		yes	47 (60)	18 (51)	29 (66)
		no	32 (40)	17 (49)	15 (34)
		missing	0	0	0
**type of iron intake**		orally	22 (53)	8 (44)	14 (48)
		i.v.	25 (47)	10 (66)	15 (52)
		missing	32	17	15
**smoking habits**		smoker	12 (17)	8 (26)	4 (10)
		ex-smoker	18 (26)	10 (32)	8 (21)
		non-smoker	40 (57)	13 (42)	27 (69)
		missing	9	4	5
**pack years**		number of years	8 ± 14	13 ± 17	4 ± 9
		missing	23	11	12
**abscess**		yes	6 (8)	4 (11)	2 (5)
		no	73 (92)	31 (89)	42 (95)
		missing	0	0	0
		yes	7 (9)	1 (3)	6 (14)
**stroke/embolism**		no	72 (91)	34 (97)	38 (86)
		missing	0	0	0
**CIDT**		yes	18 (23)	8 (23)	10 (23)
		no	61 (77)	27 (77)	34 (77)
		missing	0	0	0

HHT = hereditary hemorrhagic telangiectasia, n = number of patients, FH = family history of HHT, TAE = telangiectasia, GI = gastrointestinal involvement, PAVM = pulmonary arteriovenous malformation, HVM = hepatic vascular malformation, CVM = cerebral vascular malformation, ESS = Epistaxis Severity Score (on a scale from 1 = mild epistaxis to 10 = severe epistaxis), course of the disease/epistaxis: patients were asked to state whether they felt the course of the disease in general/their epistaxis had worsened/was equal/got better over the last two months, type of iron intake: orally = patients who took tablets or juice, i.v. = patients who also took iron supplements intravenously (percentages are calculated out of all patients who took iron supplements), CIDT = patients with HHT and chronic inflammatory or immunosuppressive diseases/treatments (n = 18): asthma (n = 3), chronic obstructive pulmonary disease (COPD, n = 4), gastritis (n = 2), Hashimoto’s thyroiditis/immunthyreoiditis (n = 3), chronic hepatitis (n = 2), therapy with immunosuppressive drugs (tacrolimus, thalidomide, n = 2), polyserositis (n = 1), lupus erythematosus (n = 2), spondylodiscitis (n = 1), psoriasis (n = 1), pyelonephritis/recurrent urinary tract infection (n = 2), rheumatoid arthritis/arthrosis (n = 1), psoriasis (n = 1), multiple selections were possible.

**Table 2 jcm-09-00767-t002:** Immune status of healthy and HHT patients of comparable age.

		Healthy (n = 23)	HHT (n = 58)	*p*
**Age**	years	50 (36–53)	53 (48–59)	*0.08*
**Gender, male**	n, %	8 (35%)	24 (41%)	*0.68*
**Current smoking**	n, %	5 (22%)	10 (17%)	*0.88*
**WBC**	10^6^/mL	6.6 (5.7–8.6)	5.7 (4.6–6.7)	*0.10*
**Neutrophils**	**% WBC** 10^6^/mL	**57 (51–68)** 3.8 (2.9–6.0)	**63 (57–72)** 3.4 (2.6–4.5)	***0.03*** *0.42*
**Monocytes**	**% WBC** 10^6^/mL	**5.5 (4.3–7)** 0.43 (0.32–0.50)	**6.8 (5.3–8.6)** 0.38 (0.32–0.47)	***0.03*** *0.51*
**Lymphocytes**	**% WBC** **10^6^/mL**	**32 (25–38)** **1.9 (1.4–2.1)**	**22 (17–28)** **1.3 (0.9–1.7)**	***0.003*** ***0.02***
**B cells**	**% lym** 10^6^/mL	**14 (11–16)** 0.30 (0.18–0.38)	**17 (13–25)** 0.21 (0.13–0.33)	***0.04*** *0.23*
**NK cells**	**% lym** 10^6^/mL	**12 (10–14)** 0.19 (0.14–0.23)	**17 (14–21)** 0.22 (0.14–0.28)	***0.002*** *0.52*
**T cells**	**% lym** **10^6^/mL**	**70 (66–77)** **1.3 (0.8–1.6)**	**61 (51–68)** **0.7 (0.5–1.1)**	***0.0008*** ***0.03***
**CD4+**	% T cells **10^6^/mL**	76 (69–82) **0.88 (0.41–1.14)**	76 (66–81) **0.47 (0.35–0.70)**	*0.72* ***0.03***
**Central memory**	**% T cells**	**32 (25–36)**	**36 (32–40)**	***0.03***
**Effector memory**	% T cells	20 (19–27)	26 (20–32)	*0.18*
**Naive**	**% T cells**	**42 (39–48)**	**30 (25–41)**	***0.01***
**Effector**	% T cells	3.0 (2.4–4.3)	3.0 (2.2–6.0)	*1.00*
**PD1+**	**% CD4+** % CD8+	**2.7 (2.0-3.0)** 2.9 (1.8–3.6)	**3.9 (2.6–5.7)** 3.8 (1.8–6)	***0.01*** *0.32*
**Plasma TGFß**	pg/mL	13 (11–15)	12 (11–13)	*0.10*

Different immune cells of patients with HHT and controls aged between 20 and 65 years were analyzed. HHT = hereditary hemorrhagic telangiectasia, n = number of patients, WBC = white blood cells, lym = lymphocytes, PD1+ = programmed cell death protein 1, Plasma TGFß = transforming growth factor-beta in plasma. Data are presented as median (25–75 percentiles) or n (%); Mann–Whitney U test was used for comparison of two independent groups, Chi-square test was used for comparison of frequencies. Significant correlations are marked in **bold** (with *p* < 0.05).

**Table 3 jcm-09-00767-t003:** Correlation of immune cell counts with red blood parameters in HHT.

	Erythrocytes /pL		Hemoglobing /dL	Hematocrit L/L	Iron µ/dL
**WBC**	10^6^/mL	**0.24**	**0.24**	0.23	**0.26**
**Neutrophils**	% WBC 10^6^/mL	−0.20 0.19	−0.18 0.22	−0.10 0.22	0.02 0.32
**Monocytes**	% WBC 10^6^/mL	−0.19 0.06	**−0.28** −0.00	**−0.26** −0.03	−0.21 0.14
**Lymphocytes**	% WBC **10^6^/mL**	0.22 **0.33**	0.21 **0.38**	0.13 **0.28**	−0.03 0.14
**B cells**	% lym **10^6^/mL**	0.12 **0.35**	−0.01 **0.33**	0.04 **0.29**	−0.20 −0.00
**NK cells**	% lym 10^6^/mL	−0.21 0.15	−0.17 0.23	−0.12 0.22	0.01 0.21
**T cells**	% lym **10^6^/mL**	0.03 **0.30**	0.13 **0.39**	0.05 **0.28**	0.16 0.20
**CD4+**	% T cells **10^6^/mL**	0.09 **0.27**	−0.05 0.26	−0.05 0.16	−0.20 −0.02

HHT = hereditary hemorrhagic telangiectasia, WBC = white blood cells, lym = lymphocytes. Spearman rank R test was used to estimate correlations between parameters, data are presented as R coefficient, significant correlations are marked in **bold** (with *p* < 0.05), number of patients with HHT = 40.

**Table 4 jcm-09-00767-t004:** Characteristics of HHT patients in relation to the type of iron-replacing therapy.

	1. No (n = 29)		2. Oral (n = 22)	3. I.v. (n = 21)	*p*
**Erythrocytes**	**/pL**	**4.8 (4.5–5.1)**	**4.5 (4.0–4.8)**	**4.2 (3.6–4.5)**	**p_1/2_ = 0.04** **p_1/3_ = 0.0003**
**Hemoglobin**	**g/dL**	**13.1 (10.6–14.5)**	**12.2 (11.2–13.3)**	**9.7 (8.7–11.7)**	**p_2/3_ = 0.008** **p_1/3_ = 0.004**
**Hematocrit**	**L/L**	**0.40 (0.35–0.43)**	**0.37 (0.36–0.40)**	**0.31 (0.30–0.36)**	**p_2/3_ = 0.02** **p_1/3_= 0.004**
**MCHC**	g/dL	33 (31–34)	33 (32–34)	31 (30–33)	0.08
**Plasma iron**	µg/dL	53 (20–78)	35 (25–62)	27 (16–42)	0.18
**WBC**	10^6^/mL	5.8 (4.8–7.0)	5.0 (4.5–5.7)	5.7 (4.6–6.8)	0.23
**Neutrophils**	10^6^/mL	3.8 (2.8–4.4)	2.8 (2.4–4.4)	3.4 (2.7–4.6)	0.48
**Monocytes**	10^6^/mL	0.39 (0.32–0.57)	0.39 (0.32–0.50)	0.36 (0.26–0.54)	0.73
**Lymphocytes**	10^6^/mL	1.4 (0.9–1.8)	1.2 (0.9–1.6)	1.2 (0.8–1.6)	0.56
**T cells**	10^6^/mL	0.74 (0.49–1.22)	0.71 (0.58–1.09)	0.63 (0.49–0.87)	0.66

The type of iron replacement was documented: no = no iron replacement, oral = iron tablet intake, i.v. = intravenous +/− oral iron intake. HHT = hereditary hemorrhagic telangiectasia, n = number of patients, WBC = white blood cells. Data are presented as median (25–75 percentiles); Kruskal–Wallis ANOVA test and Mann–Whitney U test were used to compare multiple and two independent groups, respectively. Significant correlations are marked in **bold** (with *p* < 0.05).

**Table 5 jcm-09-00767-t005:** Age and gender of HHT patients and healthy donors analyzed for reactive oxygen species (ROS) production and neutrophil extracellular trap (NET) release.

		Healthy (n = 7)	HHT (n = 9)	*p*
**Age**	years	55 (42–62)	52 (45–67)	*0.87*
**Gender, male**	n (%)	2 (29%)	3 (33%)	*0.73*

HHT = hereditary hemorrhagic telangiectasia, n = number of patients. Data are presented as median (25–75 percentiles) or n (%). Mann–Whitney U test was used for comparison of two independent groups. Chi-square test was used for comparison of frequencies.

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
