# Peer review of "Impaired Release of Neutrophil Extracellular Traps and Anemia-Associated T Cell Deficiency in Hereditary Hemorrhagic Telangiectasia"

_jcm, 2020, doi:10.3390/jcm9030767_

Round 1

Reviewer 1 Report

Thank you very much for letting me review the manuscript „Release of neutrophil extracellular traps is impaired in hereditary hemorrhagic telangiectasia“ by Droege and colleagues.

This study is registered with clinicaltrials.gov (NCT02983253)

The authors prospectively assessed alterations of the immune system in 79 patients with hereditary hemorrhagic telangiectasia, in comparison to 45 healthy controls, to reveal factors that could predispose HHT patients to immunosuppression and recurrent bacterial infections.

The study found a quantitative decrease of lymphocytes in absolute numbers and qualitative impairments of neutrophils including decreased capacity to produce NETs (which is the major novel aspect of the work), impaired cell adhesion and migration in patients with HHT.

The authors conclude that reduced NET formation and resulting insufficient NET-mediated bacteria killing may contribute to the impaired bacterial clearance in HHT patients and suggest that stimulation of neutrophil antimicrobial activity may have the potential to serve as a novel therapeutic approach to reduce susceptibility to infections.

In recent years, neutrophil extracellular traps have been recognized as a central player in antimicrobial host defense and inflammation. NET components may serve as therapeutic target structures and are thus of great interest in several medical conditions. Little is known about the regulation of immune responses and the pathophysiologic role of neutrophils and their mediators in HHT patients. In particular, no data are yet available on NETosis in HHT.

Overall, the manuscript is well written and the topic is interesting, the methodology is appropriate.

The abstract may be improved and should clearly state the main methods and the main results of the study (i.e. the observed differences of NET production between patients and controls with median, IQR and p value) as well as the number of patient samples in which anti-bacterial activity of neutrophils was assessed. Antimicrobial activity (ROS production and NETosis) and structural changes in neutrophil cytoskeleton were assessed in only 9 patient samples (and 7 healthy volunteer samples). This should be mentioned in both the abstract and the limitations section. Given that the study evaluated a variety of immunologic parameters and that bacteria-triggered NETosis was assessed in a rather small number of samples, the authors may consider rephrasing the study title.

The introduction is straight and provides sufficient background. The hypothesis and the aim of the study are clear and well described. Study findings should not be part of the background section (P2 L64-68) - the authors may consider removing these two sentences from the Introduction. Multivariate regression analysis (P15 L310) is not mentioned in the Methods section. Potential clinical implications of the study findings (in particular therapeutic strategies to restore neutrophil antibacterial functions in HHT patients) should be discussed in the light of study results and whole evidence.

Virtues and limitations of the study should be stated clearly in a separate Limitations section. Include recommendations for the future course of action.

Reviewer 2 Report

The authors investigate blood parameters, and more specifically neutrophil extracellular traps, in patients with hereditary hemorrhagic telangiectasia. Compared to age-matched healthy volunteers, HHT patients had reduced lymphocytes. Correlation analyses showed significant negative correlations between lymphocytes and age, and RBC parameters and age, only in HHT patients. They also examine NET formation and ROS production in these patients. There are several issues to be addressed in order for the conclusions in the manuscript to be supported by the data as presented:

1. The link between the correlation analyses for lymphocytes/monocytes/RBCs/epistaxis and the main point of the paper: the effect on neutrophils, is not clear. Were neutrophils also significantly correlated with age of epistaxis severity? 

2. Why are monocytes shown as %WBC in the correlation analyses, while all other parameters are absolute counts?

3. The conclusion in the text is that blood loss is responsible for the loss of lymphocytes. Why would blood loss lead to preferential loss of this cell type and not of neutrophils/monocytes (especially neutrophils since these are the most prevalent leukocytes in circulation)?

4. The presence of a subgroup of HHT patients who are taking immunosuppressive medication is a confounding factor in this study. What types of medications were taken? Were CIDT patients included in the in vitro NETs analyses?

5. Is the TGF measured total TGFb or the active form? 

6. Page 18 line 372 - text says "slightly reduced in HHT" but the data does not show statistical significance. Please reword.

7. Why was P. aeruginosa chosen as the bacterial stimulus and not a strong NET inducer such as S. aureus, which the authors refer to as a common infection in HHT patients? Does P. aeruginosa induce significant NET formation compared to control in healthy patients? In HHT patients? These analyses are not presented.

8. Lastly, the NET quantification does not seem to reflect the representative images shown. Mean area of NETs does not necessarily indicate the percentage of cells which have formed NETs, but rather the total area covered by (any) NETs. The standard error in control wells healthy volunteers is very high, and does not instill much confidence that the NET assay provides reliable information as to NET formation potential. I would recommend that this analysis be revisited to reflect percentage of NET-forming neutrophils. 

9. P = 0.07 in Figure 5E does not support the wording in the text of "better migratory capacity" in the figure legend. 

10. The phalloidin experiments are performed only in a very short-term manner (15 minutes, before any NETs would be formed). F-actin disassembly is observed at later stages in NET formation. Is there any effect seen at later time points?
